# The Role of Microencapsulation in Food Application

**DOI:** 10.3390/molecules27051499

**Published:** 2022-02-23

**Authors:** Mariel Calderón-Oliver, Edith Ponce-Alquicira

**Affiliations:** 1Tecnologico de Monterrey, Escuela de Ingeniería y Ciencias, Avenida Eduardo Monroy Cárdenas 2000, San Antonio Buenavista, Toluca 50110, Mexico; 2Departamento de Biotecnología, Universidad Autónoma Metropolitana, Unidad Iztapalapa, Avenida San Rafael Atlixco 186, Col. Vicentina, Mexico City 09340, Mexico; pae@xanum.uam.mx

**Keywords:** microencapsulation, fat substitute, sensory improvement, functional food, preservatives

## Abstract

Modern microencapsulation techniques are employed to protect active molecules or substances such as vitamins, pigments, antimicrobials, and flavorings, among others, from the environment. Microencapsulation offers advantages such as facilitating handling and control of the release and solubilization of active substances, thus offering a great area for food science and processing development. For instance, the development of functional food products, fat reduction, sensory improvement, preservation, and other areas may involve the use of microcapsules in various food matrices such as meat products, dairy products, cereals, and fruits, as well as in their derivatives, with good results. The versatility of applications arises from the diversity of techniques and materials used in the process of microencapsulation. The objective of this review is to report the state of the art in the application and evaluation of microcapsules in various food matrices, as a one-microcapsule-core system may offer different results according to the medium in which it is used. The inclusion of microcapsules produces functional products that include probiotics and prebiotics, as well as antioxidants, fatty acids, and minerals. Our main finding was that the microencapsulation of polyphenolic extracts, bacteriocins, and other natural antimicrobials from various sources that inhibit microbial growth could be used for food preservation. Finally, in terms of sensory aspects, microcapsules that mimic fat can function as fat replacers, reducing the textural changes in the product as well as ensuring flavor stability.

## 1. Introduction

The use of microparticles is a worldwide trend that is constantly expanding in various areas such as medicine, food, electronics, and environmental remediation, among others, as microparticles can carry and protect several active compounds with broad applications. For instance, the incorporation of microcapsules allows for the development of materials with better mechanical and functional properties; in medicine, microcapsules function as vehicles for the targeted administration of specific chemicals or sensors. In general, microcapsules are applied for containing bioactive compounds and protecting them from humidity, oxygen, light, and some other environmental factors. Microencapsulation acts as a barrier to control release, solubility, and bioavailability; facilitates handling and transport; and can also mask unpleasant flavors and aromas [1,2].

Microcapsules usually range between 0.2 to 5000 μm in diameter and consist of an encapsulating or wall-material that englobes a core containing the active substance. The final particle size depends on many factors, such as the processing method and the nature of the encapsulating material [3]. Therefore, it is important to consider the type of wall-material that will be used in combination with a specific encapsulation process according to the function or destination of the microcapsule and the desired particle size; the wall material also leads to variation in encapsulation efficiency and stability [4]. Table 1 summarizes the various techniques used for the preparation of microcapsules and Table 2 presents some application techniques as well as some of the wall and core materials that have been used in foods. An interesting aspect of the application of microcapsules in food—and especially of the material and the technique with which they are conducted—is that in some cases, the microcapsules serve to conserve a compound or bioactive extract in the product, controlling its release during storage; however, in other cases, microcapsules promote the product release in vivo during digestion. For example, microcapsules of carboxymethylcellulose and chitosan produced by the coacervation method allow the gradual release of carotenoids; while microparticles made with the same technique but with different wall-materials (chitosan–sodium tripolyphosphate) release the carotenoids much more quickly (25%) in both yogurt and bread [5].

Microcapsules in foods have various functionalities, which are summarized throughout this review of publications from the last 6 years (2015–2021) listed in databases such as SCOPUS, PubMed, and Web of Science. The search included words such as “microencapsulation” plus “meat”, “milk”, “bread”, and “juice”, among other foods considered in the review. Articles were considered in which microcapsules were added during food preparation and presented accordingly with their specific functionality or application as a preservative, functional food development, sensory improvement, fat substitution, and/or others, which will be broken down throughout the review (Figure 1).

## 2. Microcapsules in Food Preservation

Preservatives are compounds that prevent spoilage in food caused by enzymes, microorganisms, oxygen, and others. They should not be toxic or modify the taste, in addition to presenting stability and effectiveness in foods to which they are added [22]. However, many of these preservatives do not meet these requirements; they can be very effective in the product, but because of the concentrations used, they can produce unpleasant flavors or exhibit decreased effectiveness when in contact with the product. Microencapsulation is an option to limit or prevent these changes. Among the most commonly encapsulated preservatives are polyphenols (such as flavonoids and tannins capable of exerting antioxidant activity and inhibiting microbial growth); organic acids (which can change the pH); essential oil extracts derived from herbs, spices, onion, garlic, and fruits (set of a variety of compounds including phenolics and organosulfur compounds with antioxidant and antimicrobial capacity); bacteriocins (peptides with antimicrobial capacity); and even phage (viruses that infect bacteria). Compounds with antioxidant capacity prevent or retard the formation of free radicals and, therefore, oxidation reactions; meanwhile, antimicrobials, through various mechanisms, are capable of disrupting membranes or cell walls and dissipating the proton motive force, inhibiting microbial growth [23]. It should be mentioned that many of the extracts with relevant properties are not soluble or compatible with the matrix to which they are to be added; therefore, microencapsulation may be relevant for their application.

### 2.1. Microcapsules Containing Antimicrobials

#### 2.1.1. Meat

The benefit of microencapsulation is to enhance and prolong the effect of a bioactive compound, even if it is subjected to long periods of heating [24]. For example, it is known that clove and its derivatives have an antimicrobial effect due to the presence of eugenol and other phenylpropanoids, which are very susceptible to light and temperature; further, their solubility and volatility may limit their possible incorporation in some food matrices. Encapsulation may improve their stability and applicability so that they can be used in various meat products. By microencapsulating clove oil in beta-cyclodextrin and starch, its fungicidal effect is enhanced by 0.08% [25]. Moreover, the microencapsulation of eugenol alone in quinoa protein and gum Arabic by complex coacervation reduces total bacterial count (from 8 to 6.5 log CFU/g) in minced meat when it is stored for 15 days at 4 °C, showing better protection against free eugenol treatment [9].

Another antimicrobial example is the microencapsulation of thymol oil (4.9 mg/g) in beta-cyclodextrin, which reduces by 0.05 the log CFU of total viable count bacteria when they are incorporated into ground meat and stored for 8 days at 4 °C [26]. In the same product, the addition of microcapsules containing *Allium sativum* essential oil (20%), made by complex coacervation with gum Arabic and maltodextrin, reduced the final count of *E. coli* (from 4.0 ± 0.5 to less than 1.0 ± 0.1 log CFU/g), total aerobic mesophilic bacteria (from 8.5 ± 0.3 to 6.4 ± 0.4 log CFU/g), and sulfite-reducing anaerobes (from 3.5 ± 0.08 to less than 1.0 ± 0.1 log CFU/g) in meat stored at 8 °C for 6 days [10].

The benefits of microencapsulation are not exclusive to polyphenols and extracts from fruits, peels, or fruit seeds but also extend to antimicrobial peptides such as nisin, whose microencapsulation reduces the growth of *Listeria monocytogenes* by up to 75% with respect to the control and is 50% more effective than nisin without encapsulation when a ham is inoculated with 10^3^ CFU/g of *Listeria* [27].

The use of specific phage for *Salmonella* Enteritidis and *Salmonella* Typhimurium in raw chicken meat at a concentration of 10^8^ PFU/g (plaque-forming unit/g) (microencapsulated in whey protein isolate and trehalose) decreases microbial growth by 0.57 log CFU/cm^2^ for *S*. Enteritidis and 1.78 log CFU/cm^2^ for *S.* Typhimurium during storage at 4 °C for 4 days without altering consumer acceptability or sensory aspects [28].

The microencapsulation of various antimicrobials has a synergistic effect with other preservation techniques, such as irradiation. An example of this is the combination of antimicrobials such as cinnamon (*Cinnamomum cassia*) and nisin microencapsulated in alginate and cellulose, which, together with 1.5 kGy irradiation, decrease the growth of *Listeria monocytogenes* inoculated in ham by 86.95% after 28 days of storage [29].

#### 2.1.2. Milk and Derivates

The inclusion of bacteria in microcapsules is not used exclusively for the elaboration of functional products, as presented below, but also for inhibiting the growth of other microorganisms and as a source of preservatives. Such is the case of the bacteria *Bifidobacterium animalis* and *Lactobacillus acidophilus* in cheese, which inhibit the growth of fungi such as *Aspergillus niger* in feta cheese during 45 days of storage; this effect is enhanced when they are encapsulated in sodium alginate, which increases the final count of viable bacterial cells by approximately 0.1 log CFU during storage [30].

### 2.2. Microcapsules Containing Antioxidants

#### Meat

Antioxidants are compounds that delay the oxidation of proteins and lipids by the addition of hydrogen atoms or electrons or the removal of oxygen atoms—examples include enzymes and polyphenols, among others.

Microencapsulation extends the effectiveness and stability of polyphenols; this has been exemplified in the application of emulsified microcapsules of mulberry polyphenol with gum Arabic in dried minced pork slices (8.5%), which, in addition to reducing oxidation reactions, also improved the color compared with nonencapsulated mulberry polyphenol, indicating that anthocyanins and other compounds in the extract were protected from the light and heat treatments to which they were subjected during meat processing [24]. Another example is the use of propolis coproduct extract containing coumaric acid and epicatechin, which function as antioxidants and, when encapsulated by the spray drying method and added to the system at a concentration of 0.3 g/kg, can inhibit the lipid oxidation of hamburger meat for 28 days at −15 °C [31]. Even the addition of encapsulated antioxidants can stabilize changes in texture resulting from protein oxidation, such as the case of the microencapsulation of procyanidins by extruding starch added to chicken sausages, inhibiting the formation of disulfide bonds in proteins, protein oxidation, and general amino acid stability on the final product [32].

Many of the rapidly growing food preservation techniques, including nonthermal treatments such as high-pressure treatment, can accelerate oxidation processes in meat. It has been observed that the addition of antioxidants prevents or delays this process; if encapsulated, they provide a better antioxidant effect. An example is the case of encapsulation of the pitaya shell extract (*Hylocereus costaricensis*) rich in polyphenols (concentration of 49.5 mg/g) when spray-drying maltodextrin, which, when added (100 and 1000 ppm) to ground pork patties subjected to high pressures (500 MPa for 10 min), decreases protein oxidation (by 25%) and prevents the changes in hardness and chewiness that occur during the process [33]. A similar case was observed with ultraviolet radiation, which induces lipid and protein oxidation; the addition of microcapsules with different concentrations of pitaya extracts (100–2000 ppm) in refrigerated ground pork patties for 10 days reduced those changes [34].

## 3. Microcapsules in Functional Foods

A functional food is one that, in addition to providing nutrients and energy to the body, can promote one or more beneficial functions in the body, either for disease prevention or to activate a physiological response [35].

### 3.1. Incorporation of Fatty Acids

Most of the compounds that are used to develop functional foods involve the addition of polyunsaturated fatty acids (especially omega 6 and 9), which are susceptible to oxidation reactions and/or change the sensory profile of the product. Fatty acids are important in food because they have been associated with cardioprotective effects such as lowering LDL (“bad”) cholesterol and increasing HDL (“good”) cholesterol, thus reducing heart attacks; in some other cases, they have been associated with cancer prevention [36].

#### 3.1.1. Meat

The microencapsulation of omega-3, -6, and -9 fatty acids improve their stability by reducing oxidation and extending shelf life as well as providing added nutritional value to the product in which they are added [37]. Proof of this is the addition of fish oil with a high content of omega-3 fatty acids, such as eicosapentaenoic acid and docosahexaenoic acid to chicken nuggets. When encapsulated by the multilayered emulsion technique followed by spray drying (150 mg of acid/100 g), the fish oil is no longer detected at the sensory level, and the changes in the appearance and textural characteristics that the oil presents when it is not encapsulated are not observed; in addition, no rancidity occurs during product storage [38], even if these microcapsules are applied in pre-fried products or frozen products [39]. However, it is important to mention that adding higher fatty acid content to a product, even if microencapsulated, causes greater susceptibility to the oxidation process [40]; therefore, it is important to consider the encapsulation method and the material to be encapsulated.

#### 3.1.2. Milk and Derivates

The microencapsulation of fatty acids helps to increase the nutritional value of dairy products. By incorporating 3.1% of omega-3 microcapsules in whole milk powder, it remains stable for longer, independent of the packaging (in a metallic tin or flexible plastic) or the storage conditions (43 °C without controlled relative humidity or 34 °C with 83% humidity) to which the product is subjected [41]. The microcapsules also help fortify dairy products, even those with partial milk replacement (30%). Adding microcapsules with oleic acid (1.5 g/100 g of microcapsules) does not modify the microstructure of the fermented milk product; although, due to the type of material that comprises the microcapsules, they can modify other properties such as the final viscosity [42].

Another example is the incorporation of microencapsulated calcium carbonate (1% in the formulation) in casein and maltodextrin into yogurt mousse, which, together with the incorporation of 4.8% inulin, develops a product with sensory and rheological characteristics similar to those of the original product, without microcapsules and without fat reduction [43].

#### 3.1.3. Cereals and Derivates

Incorporation of microcapsules with 5% fish oil (omega-3 fatty acids) made from chitosan and modified starch in bread does not modify the acceptability of the product, as it increases firmness (46%) and color in the a* and b* scales (152 and 8%, respectively). The nutritional value of the product was shown to increase [44] and the formation of hydroperoxides was prevented in the product by up to 70% relative to the oil without encapsulation [45]. In addition to bread, fish oil encapsulated in milk by the spray drying method has also been added to cookies, which, in addition to modifying the nutritional value, decreases the oxidation reactions of the product by 40% with respect to the oil without encapsulation [46]. Another example in bread involves the incorporation of flaxseed oil encapsulated in yeast cells, which, in addition to fortifying the bread, prevents the oxidation of fatty acids [47].

#### 3.1.4. Fruit and Juices

The fortification of juices with fatty acids has also been studied by incorporating fish oil into microcapsules made by complex coacervation, which increases turbidity but does not make the juice sensorially unacceptable [8].

### 3.2. Microcapsules Containg Prebiotics and Probiotics

Probiotics refers to beneficial microorganisms within the intestinal microbiota and have high beneficial potential in host health; while prebiotics (mostly) are indigestible oligosaccharides that benefit the host by stimulating the growth and activation of bacterial metabolism of the microbiota [48]. However, upon incorporation into a food, probiotic growth decreases because the bacteria are not in the ideal growth conditions. Probiotics and prebiotics can modify the sensory characteristics of the product and microencapsulating them helps to reduce these changes.

#### 3.2.1. Meat

The alginate and pectin microencapsulation of lactic bacteria, together with prebiotics such as pear cactus peel flour and 1% apple marc flour, increase the number of beneficial bacteria (between 165 and 185%) and decrease the number of pathogens by 100% when they are added at 5% to cooked sausages for 15 days of storage [49]. Microcapsules with probiotics have even been studied in fermented products and found to not alter the sensory properties of the product or the fermentation process but maintain the viability of the bacteria and fulfill the goal of reaching the consumer [17]. Another example is the addition of *L. plantarum* microencapsulated by the spray drying technique on salami, where the final count is higher than 8 log CFU/g and the microcapsules did not influence the sensory acceptance [50].

#### 3.2.2. Milk and Derivates

The largest number of functional products have been characterized and implemented in yogurts because yogurt is a product that typically presents the ideal growth conditions for lactic bacteria, in addition to its compatibility with the use of various encapsulation techniques in the product [51,52,53,54].

Microencapsulated probiotic bacteria such as *Bifidobacterium* BB-12 in reconstituted goat’s milk and inulin obtained by the spray drying method increases the antagonistic effect against pathogenic bacteria such as *E. coli* in comparison with the unencapsulated probiotic as the microencapsulation simulates bowel conditions [55]. Part of this protective effect is because the microencapsulation process keeps the bacteria viable; for example, in quince seeds, *L. rhamnosus* bacteria remain viable up to 43.8% after 21 days under gastrointestinal simulation conditions in a milk dessert and *Lactobacillus* remains viable in yogurt for 180 days at −20 °C [56,57]. In some cases, the microencapsulation of *Lactobacillus casei Shirota* in gum Arabic even increases the bacteria count during storage; for example, applied in a pudding for 14 days in refrigeration increased bacteria from 8.27 to 9.16 log CFU/g and in chocolate milk for 180 days at 25 °C increased to >8 log CFU/g [58]. In pure milk, the encapsulation of *B. bifidum* BB01 bacteria in xanthan gum and chitosan increases by 0.5 log CFU/g over 21 days of milk storage at 4 °C [59]. In mature cheeses of the gouda type, a count of 10^8^ CFU/g of *Bifidobacterium lactis* is maintained for 40 days after microencapsulation in cyclodextrin and gum Arabic, and the microcapsule also adds fiber (1%) [60]. An interesting detail about adding microcapsules with probiotics to food is that they do not have adverse effects on weight gain, hematological parameters, and vital organ function in mice when they are consumed and fulfill their function when colonizing the intestine [58].

#### 3.2.3. Cereals and Derivates

As in meat and dairy products, microencapsulation is a good source for the incorporation of probiotics and prebiotics in bread. For example, alginate and starch beads coated with chitosan maintain the viability of *L. acidophilus* and *L. casei* 4 days after baking in hamburger buns and white pan bread without altering the sensory characteristics of the product [61].

#### 3.2.4. Fruits and Juices

As in the other products, microencapsulation serves as a carrier of probiotic microorganisms to generate functional products. However, in juices, this process can be more difficult due to the low pH values, which can damage the microcapsules during the storage period or affect their stability and, therefore, that of the microorganisms. Thus, protection in juices is different from that in other products. For example, juices such as pineapple, raspberry, and orange—whose pH values, are 3.28, 2.75, and 3.45, respectively—degrade the microcapsules in various proportions, with raspberry being the juice that decreased the number of microcapsules and the number of viable microorganisms the most [62]. These juices even lower the pH of the product to which they are added [63]. However, this problem has been solved by incorporating natural extracts such as moringa and green tea into the microcapsules containing the microorganisms. This promotes viability upon addition to fruit juice (kiwi, prickly pear, and carrot) and to yogurt and increases the stability of the microcapsules and viability of microorganisms with the incorporation of oligofructose or other prebiotics, such as inulin and fructo-oligosaccharides [12,13,14]. In other studies, both fruit juice and probiotics are incorporated into the microcapsule, which gives it more stability and allows the development of functional powdered beverages [64]. In addition to providing protection and viability, the microencapsulation of bacteria promotes greater digestion (in simulated conditions) and improves the sensory acceptability of the juice [65].

The incorporation of various types of microcapsules in juices resists pasteurization processes well. However, researchers are beginning to explore the effects of nonthermal treatments, such as high pressures. It has been observed that bacteria that are subjected to high pressures are more resistant to stress and survival [66]. This viability can be increased further if they are encapsulated. When added to mandarin juice, there are changes in the physicochemical properties [67]; therefore, this is a good area of opportunity for the application of microcapsules. The protection of probiotics during the fermentation process is found in the case of apple juice, where the encapsulation of *Lactobacillus plantarum* in alginate maintains the viability at 2 log CFU/mL more than the unencapsulated treatment [68]. 

#### 3.2.5. Other Products

Within the development of new products, and in favor of healthier food consumption trends, functional beers have been developed (5% vt alcohol content) that contain microencapsulated probiotics (*Lactobacillus rhamnosus* GG in alginate and silica-coated alginate) [19]. 

### 3.3. Microcapsules with Antioxidants in Functional Products

Antioxidants can serve a double function: to preserve the product or to have a beneficial effect on the consumer when a certain antioxidant content remains in the finished product. Antioxidants decrease the presence of free radicals and other compounds that are associated with various diseases where oxidative stress is present; therefore, they can prevent or counteract the side effects of metabolic syndrome and other diseases [69].

#### 3.3.1. Milk and Derivates

The incorporation of polyphenols as a source of antioxidants for the consumer can provide undesirable flavors or shorten their shelf life when interacting with light or with the components of the food matrix in which they are incorporated. The microencapsulation of catechin, a polyphenol present in several products, such as green tea, grapes, and cocoa, has been reported to have health benefits. When encapsulated in cyclodextrin and added to yogurt and milk (at 0.1 mg/mL), its flavor is masked without affecting other sensory parameters, and 82% of the initial catechin is preserved after a simulated digestion process [70].

#### 3.3.2. Cereals and Derivates

As in milk, microencapsulation helps maintain stable health compounds, such as anthocyanins, even after a thermal process, such as cookie baking [71]. Such is the case regarding the incorporation of microcapsules of sour cherry pomace extract in whey protein, which prevents the loss of anthocyanins due to processing and preserves the antioxidant capacity of the extract in the product [72]. Similarly, the microencapsulation of black rice extract (*Oryza sativa* L.) in biscuits keeps 50% more polyphenols in the product than does the control [6]. Similar behavior is observed with the microencapsulation of the beetroot pulp network (a byproduct of the juice industry), which contains betalains that can be incorporated into water biscuits, increasing their antioxidant capacity as well as decreasing the concentration of furosine (a molecule that decreases protein digestibility) by 50% [73]. The incorporation of microencapsulated green tea polyphenols into maltodextrin made by the lyophilization method maintains a considerable content of polyphenols after the bread-baking process, up to 62.84% [74], and even preserves compounds with beneficial health effects, such as hydroxy-citric acid, during the baking process [7].

### 3.4. Other Functional Benefits of Microcapsules

#### 3.4.1. Milk and Derivates

One of the most common reasons why people do not consume dairy products is because they lack the enzyme lactase; thus, they must consume this enzyme externally or consume lactose-free products. Microencapsulation has proposed a solution to this problem by incorporating the enzyme into the same dairy product. β-D-galactosidase was microencapsulated in hydroxypropyl methylcellulose phthalate, which was added to milk (approximately 0.148 g of enzyme), controlling its release, preventing its hydrolysis, and retaining 81.18% of the enzyme after 12 days in refrigeration [75]; the formulation of microemulsions in caseinates and sodium lecithin has also been proposed to encapsulate lactase and control its release and degree of hydrolysis in products such as skim and full-fat milk [75,76].

Another interesting fact is that the incorporation of microcapsules with minerals such as iron in infant milk formula increases their absorption compared with when the mineral is added without any protection, as the microcapsules of iron encapsulated in nondigestible polymers, such as resistant starch and pectin, increase iron absorption by up to 53% in rats fed powdered milk [77]. An effect similar to that on infant milk formula is observed in feta cheese that incorporates microcapsules of iron and ascorbic acid (80 mg/kg and 150 mg/kg, respectively), which increases the iron content compared with the unmodified product and remains within sensory parameters [78]. A similar effect is observed on milk fortification by incorporating iron microencapsulated in maltodextrin, gum Arabic, and modified starch [20].

#### 3.4.2. Cereals and Derivates

As mentioned earlier, the microencapsulation of minerals such as iron increases resistance to food processing and bioavailability in milk. This also occurs in the process of baking bread. By adding iron microcapsules to modified starch, the bioavailability is increased from 14% for iron without encapsulating to up to 99%, indicating that the microcapsules protect iron from inhibitors such as chelating agents, phytic acid, and its derivatives, and increase solubility and degradation during the digestion process [79]. This example is interesting in that the microcapsules do not interfere with the fermentation process of the product, and substances that help iron absorption, such as ascorbic acid, can also be included in the microcapsule.

## 4. Microcapsules for Fat Replacement

The type and amount of fat in food are of the utmost importance and improve the sensory characteristics of the product, such as texture, taste, and even color. In some cases, fat is the main ingredient, when part of an emulsion. Partially or totally replacing fat is a common strategy used to create products with fewer calories. This implies a change in sensory acceptance or shelf life, and the substitution may not be appropriate depending on the functionalities of the fat in the product. It has been observed that microcapsules can be a fat substitute, depending on the technique with which they are made, as it has been reported that microcapsules made by the complex coacervation technique can function as fat mimetics [11].

### Fat Replacement in Meat

The replacement of fat with empty microcapsules or with the incorporation of some bioactive compounds improves sensory appearance parameters such as texture in addition to extending the stability of the products. An example of this is the incorporation of microcapsules with chia oil and rosemary at 10% of the final formulation to substitute 50% of the amount of fat (pork back fat) that is added to hamburger meat, considerably improving sensory parameters such as the texture and positive descriptors such as pleasant herbal odors after 120 days of storage compared with the control (to which the fat was completely replaced), which presented odors resulting from the oxidation of lipids [16]. It should be mentioned that the microcapsules were made by the ionic gelation in the sodium alginate technique. Similar behavior has been observed with microcapsules that contain only chia oil when used to replace 50% of the fat in hamburger meat [80].

As seen in the functional food section, the incorporation of microcapsules with fatty acids in certain studies, in addition to increasing the nutritional value, also decreased the content of added fat. Such is the case for the addition of microcapsules of konjac glucomannan, which contain fatty acids and replace 25, 50, and 75% of the fat in a Spanish sausage by decreasing the fat content and modifying the textural properties of the product such as hardness, gumminess, and chewiness [81]. Similar results were obtained with the microencapsulation (emulsion and spray dryer with caseinate and lactose) and incorporation of vegetable oils on deer pate, which decreased the cholesterol content and balanced its fatty acid composition [37].

## 5. Application of Microcapsules for Sensory Improvement

The sensory aspects of a product are of great relevance, as they are strongly involved in the initial intention to buy a product [82]. It has been observed that the addition of various types of microcapsules can modify product texture, appearance, taste, and color, as explained in this section.

### 5.1. Meat

The microencapsulation of probiotic bacteria, in addition to adding value to the product in terms of health, also improves the aroma, sensory appearance, and other important attributes in the product. *Lactobacillus rhamnosus* microencapsulated in a mixture of polymers (alginate, gellan gum, gelatin, fructooligosaccharides, and peptides; probiotic concentration approximately 10^7^ CFU/g) using the extrusion technique and added as a starter culture in a fermented product promotes an increase in the concentration of volatile compounds with favorable sensory qualities, such as aldehydes, esters, phenols, and terpenes [17]. Not only probiotic bacteria but also the bacteria used as starter cultures remain viable through microencapsulation, which also improves the sensory properties of sausages [83].

The microencapsulation of natural dyes is a trend that helps the substitution of synthetic dyes, increasing their effectiveness and protection against environmental factors. Maltodextrin microcapsules containing jabuticaba extract (*Myrciaria cauliflora*) can be used as dye substitutes or decrease the concentration of dye required when applied at 2% in fresh sausages, in addition to having benefits as an antioxidant and antimicrobial [84].

### 5.2. Milk and Derivates

Natural dyes, such as canthaxanthin, when microencapsulated in alginate and high methoxyl pectin using the multiple emulsion and external gelation technique, protect against the color degradation that occurs in fermented dairy products when unencapsulated canthaxanthin is added [85].

Similar to color, microencapsulation allows flavor to gradually release and persist for longer compared with flavor that is not encapsulated. An example of this is the incorporation of caramel microcapsules in milk with coffee for 15 days under accelerated storage conditions at 30 °C, where the presence of volatile flavor compounds is still detected, whereas nonencapsulated caramel is no longer detected at 9 days after storage [86].

In yogurt, the incorporation of alginate–chitosan microcapsules containing *Streptococcus thermophilus* and *Lactobacillus delbrueckii* (made by the ionic extrusion-gelation method) increase product acceptance in terms of aroma and taste by consumers, a qualification that is higher with respect to the product that does not contain encapsulated bacteria, due to the generation of various organic acids [18].

### 5.3. Cereals and Derivates

The addition of microcapsules with sour cherry pomace extract in whey protein modifies the final texture of cookies, increasing the softness (by 43%) and red color (by 1400%), which could serve to color a strawberry cookie, or when a pink color is desired without using extra synthetic or natural dyes, and reduces the brightness by 33% [71].

The presence of enzymes can significantly modify the sensory profile of a product. Such is the case with the addition of the glucose oxidase enzyme in wheat flour dough and in steamed bread, which has good performance in improving bread texture and loaf volume. However, the enzyme has low stability, and its activity decreases during processing. Encapsulating it (300–1000 U/kg) decreases these effects and decreases its catalytic speed, which makes the mass present better properties such as extensibility and elastic modulus. In bread, encapsulation results in better microstructure, texture properties, and general sensory appearance. A similar effect is produced with the microencapsulation of alpha amylase in beeswax, which, when added to gluten-free bread, decreases its catalytic efficiency 2-fold and provides greater thermal stability. The product has a lower hardness and better sensory quality and acceptability [87].

The volatile compounds that provide the aroma in various products are susceptible to changes in temperature during processing or storage, in addition to contributing to one of the most important sensory parameters in the acceptability of a product. The microencapsulation of aromas provides greater sensory quality in not only aroma but also taste and global acceptance, and even increases the intention to purchase the product—for example, in a cheese bread containing Swiss cheese microcapsules in maltodextrin and starch modified and prepared by the spray drying method [88].

### 5.4. Fruits and Juices

The encapsulation of juices by various techniques to create juice powders allows the elimination of some desirable and undesirable aromas in the final product. This is exemplified by microencapsulation in maltodextrin by spray drying of fermented noni juice from which undesirable volatile compounds such as hexanoic acid and octanoic acid were removed [89].

## 6. Perspectives of Microencapsulation in Foods

As mentioned in this review, microencapsulation constitutes a great area to develop solutions in the food industry by the incorporation of active and functional ingredients within the food matrix. Microcapsules not only offer protection and enhance bioavailability and stability of bioactive substances and natural dyes, but also improve flavor by masking the taste of fortified foods with added vitamins and minerals, and prevent their interaction with other ingredients, thus facilitating their delivery. However, some constraints may be taken to verify the quality and stability of microencapsulated ingredients such as cost, encapsulation efficiency, water solubility, release rate, particle size, taste, and microscopic structure to avoid premature release, among other undesired outcomes.

Palatability plays a role for the inclusion of microcapsules in the food matrices, as they may be rejected by consumers, especially for large-sized particles or due to variations in consistency and texture. One of the key points includes the compatibility of the food matrix that must be capable of masking the presence of large-sized microcapsules. Generally, microcapsules refer to particles ranging from 0.2 to 5000 μm, while nanocapsules are smaller than 0.2 μm. Encapsulation in the nanoscale can be used to elaborate nanodispersions with greater efficiency to be used in the beverage industry, as well as for the development of packaging materials.

Furthermore, encapsulation efficiency, size, density, and stability vary according to the encapsulating process and wall material, but factors such as speed homogenization, pH, addition of mono or divalent ions (NaCl, MgCl_2_), and ionic strength must be carefully controlled to optimize the microcapsules’ properties. In addition, the microcapsule structure for single-core or multicore particles with single or multiwall structures may vary within the manufacture process.

Therefore, advances in microencapsulation involve the development of resistant and safe biopolymers that constitute a wall structure capable of preventing undesired core leakage and fulfill food industry requirements in relation to the desired application while complying with regulatory and safety aspects. Another consideration is the lack or inconsistency of the safety and toxicity studies related to the use of microcapsules; therefore, further studies must be conducted to support the legislation and safety regulations of microcapsules for food consumption.

## 7. Conclusions

The application of microcapsules in food represents a very important area of opportunity, as applications that are very common for some products have not been explored in other products, such as antioxidants in emulsions of dairy products or probiotics in bakery products. There are many articles that discuss the formation, optimization, and stability of microcapsules in various media; some of these microcapsules could be used in certain food matrices but have not been followed up in their application or in their development on an industrial scale.

As observed throughout the review, the incorporation of microcapsules in food has allowed the successful development of functional, low-fat, or sensorially improved products, among others. The diversity of techniques and materials for its preparation provides great versatility in terms of applications and functionalities in the various food products in which they are used, including resistance to heat treatments such as pasteurization and baking.

## Figures and Tables

**Figure 1 molecules-27-01499-f001:**
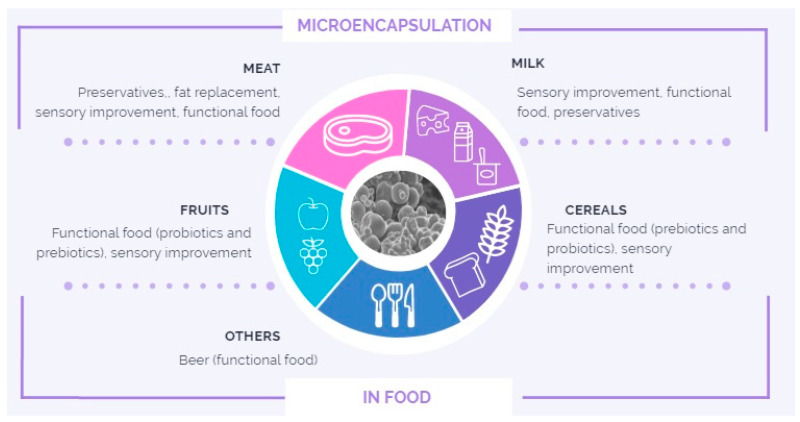
Main applications of microcapsules in food.

**Table 1 molecules-27-01499-t001:** Advantages and disadvantages of some common microencapsulation techniques.

Microencapsulation Technique	Process	Advantages	Disadvantages	Representative References
Spray drying	Drying of particles in suspension or in emulsion using hot air. The solution, emulsion or suspension is atomized in the equipment.	High production rates and efficiency Easy handling product powders Reproducibility Low operation cost It is used in a wide variety of compounds, with diverse polarities and compositions. Short time process	Not recommended for thermolabile compounds Nonuniform particles Can form aggregates	[4,6,7]
Complex coacervation	Combination of 2 polymers, such as protein and carbohydrate at specific pH value and proportion	Heat-resistant Different core compounds can be used Stable products	Different forms depending on materials Expensive Variable encapsulation efficiency Use of organic solvents	[8,9,10,11]
Encapsulation in cyclodextrins	Inclusion molecular complex in a cyclic oligosaccharide	Controlled release of actives Solubility and stability of hydrophobic actives Reduce loss and volatility of compounds	Expensive material Restricted to low-molecular-weight compounds Can form aggregates	[12,13,14]
Spray chilling	Microcapsule made up of lipids and the compound of interest, which are atomized in a cold chamber, leaving a solid particle	Low operation cost Suitable for heat-sensitive actives	Scaling parameters (melting, atomizer air temperature and pressure, cooling temperature, feed flow) Rapid release of actives Specific for hydrophobic compounds Nonuniform particles Variable encapsulation efficiency	[15,16]
Extrusion	Physical–mechanical process that involves the extrusion of the material through a nozzle	Cost-effective method No need for high temperatures, nor the use organic solvents or any specific pH condition for its elaboration.	Different sized and shaped products Difficulties with viscous solutions	[12,17,18]
Freeze drying	Freezing, sublimation (lyophilization) and desorption	Good option for temperature-sensitive compounds	Slow process Styrofoam texture Product cost	[4,7,19]

**Table 2 molecules-27-01499-t002:** Examples of microencapsulation techniques and their application in food.

Microencapsulation Technique and Conditions	Wall Material	Core Material	Food System Application	Reference
Spray drying (150 °C inlet temperature; feed flow 7 mL/min and airflow 40 m^3^/h)	Maltodextrin and gum Arabic (50:50, *w*/*w*)	Artemide black rice extract (polyphenols 122 ± 4.6 mg/g extract)	Biscuit (0.32% microcapsules in formulation, total polyphenols 975 ± 13 µg/g biscuit)	[6]
Complex coacervation (Oil/water emulsion, 50 °C, pH 4, and lyophilized)	Gelatin and gum Arabic (1:3)	Omega-3 fish rich oil	Pomegranate juice (0.04, 0.07, 0.1% powder microcapsules, i.e., 50, 100 and 150 mg DHA + EPA/L)	[8]
Modified solvent evaporation (mix solution of core and coat material, sonication at 5 °C with 5-s pulse rate for 15 min, spray in chilled alcohol, and later evaporation)	Gum Arabic, maltodextrin, modified starch (4:1:1)	Ferrous sulphate hepta hydrate	Fresh cow and buffalo milk (1:1), iron salt 25 ppm	[20]
Coextrusion (coextrusion equipment with inner (150 µm) and outer nozzles (300 µm), vibration frequency of 300 Hz, pressure 600 mbar, and voltage of 1.5 kV)	Sodium alginate (1.5% *w*/*w*) with chitosan (0.1%) and CaCl_2_ (different concentrations	*Lactobacullis plantarum* 299v and oligofructose	Ambarella juice (more than 10^7^ CFU/mL) and oligofructose (4%)	[12]
Electrospray + mineralization + freeze drying (equipment with stainless steel sterile needle and aluminium plate with collector dish, voltage 7.5 kV, flow 15 mL/h, followed by the addition of (3-aminopropyl)trimethoxysilane and tetramethyl orthosilicate) and freeze drying at −85 °C for 20 h.	Sodium alginate (1.0% *w*/*w*) and CaCl_2_ (1.5 wt%)	*Lactobacillus rhamnosus* GG	5.2 × 10^6^ CFU/mL in apple juice (pH 3.6) and 5.2 × 10^6^ CFU/mL in beer (5 vt% alcohol content)	[19]
Liposomes + spray drying. Lecithin solution at high-pressure homogenization (25,000 psi) followed by deposition of chitosan layers. Spray dryer conditions: 90 °C outlet temperature, 160 °C inlet temperature, 2.5 cm^3^/min feed rate and 0.67 m^3^/min air flow	Lecithin (2% w/w) and Chitosan (0.2%) + maltodextrin (20%) + lecithin (0.05%)	Sour cherry extract	Stirred-type yogurt (pH 4.5; 4 mg GAE/100 g)	[21]
Spray chilling + spray drying or spray drying + spray chilling (Spray dryer: 120 °C inlet air temperature and 50 °C outlet air temperature, feed rate 16.5 mL/min. For spray chilling: molten hydrogenated palm oil, homogenized, and spray chilled with nozzle fixed at 38 °C, compressed air at 0.3 bar, aspiration rate of 20 m^3^/h	Gum arabic and β-cyclodextrin (9:1 *w*/*w*) and hydrogenated palm oil	*Saccharomyces boulardii*, *Lactobacillus acidophilus*, *Bifidobacterium bifidum*	Cakes (cream-filled, marmalade-filled, and chocolate-coated; around 1 and 4.3 Log CFU/g after baking)	[15]

DHA—docosahexaenoic acid; EPA—eicosapenta-enoic acid; GAE—gallic acid equivalents.

## Data Availability

Not applicable.

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
