# Peer review of "The Role of Microencapsulation in Food Application"

_molecules, 2022, doi:10.3390/molecules27051499_

Round 1

Reviewer 1 Report

Table 1-2 are not readable between successive microcapusing technologies there should be either horizontal lines or a clearer spacing

A very thorough review of the current state of research and development directions of microencapsulation. A certain difficulty in reading is the fact of large-scale separation of chapters with the same name in the context of capillation of other substances, ingredients, etc. suggests combining into one chapter without designating - possibly the food groups discussed should be marked in italics.

Apart from the aspect directly related to the current state of knowledge, it would be reasonable to know the opinion of the authors on the discussed issue. I believe that conclusions alone are not enough for the authors to be able to express their optimism or negation about the discussed aspects.

Author Response

Response to Reviewer 1 Comments

We are very grateful for the useful suggestions raised by the reviewers and have revised the work as requested. The changes have been highlighted in red in the revised manuscript. We specify the changes below.

The review molecules-1600371 « The role of microencapsulation in food application» provided an overview of the state of art of the microencapsulation in the food industry.

Point 1. Table 1-2 are not readable between successive microcapusing technologies there should be either horizontal lines or a clearer spacing

Response 1. Spaces and lines have been corrected and are legible

Point 2. A very thorough review of the current state of research and development directions of microencapsulation. A certain difficulty in reading is the fact of large-scale separation of chapters with the same name in the context of capillation of other substances, ingredients, etc. suggests combining into one chapter without designating - possibly the food groups discussed should be marked in italics.

Response 2. The subtitles were formatted in italics. We thought that it was not feasible to group the foods since the meaning of the application of the microcapsules by function could be lost.

Point 3. Apart from the aspect directly related to the current state of knowledge, it would be reasonable to know the opinion of the authors on the discussed issue. I believe that conclusions alone are not enough for the authors to be able to express their optimism or negation about the discussed aspects.

Response 3. Excellent observation, we totally agree with it. For this reason, we added section number 6, where the perspectives are discussed. The added paragraph is attached below.

“6.       PERSPECTIVES OF MICROENCAPSULATION IN FOODS

As mentioned in this review, microencapsulation constitutes a great area to develop solutions in the food industry by incorporation of active and functional ingredients within the food matrix. Microcapsules not only offer protection and enhance bioavailability and stability of bioactive substances and natural dyes, but also improve flavor by masking the taste of fortified foods with added vitamins and minerals, and prevent their interaction with other ingredients, thus facilitating their delivery. However, some constrains may be taken to verify the quality and stability of microencapsulated ingredients such as cost, encapsulation efficiency, water solubility, release rate, particle size, taste, and microscopic structure to avoid premature release, among other undesired outcomes.

Palatability plays a role for the inclusion of microcapsules in the food matrices, as they may be rejected by consumers, especially for large-sized particles or due to variations in consistency and texture. One of the key points includes the compatibility of the food matrix that must be capable of masking the presence of large-sized microcapsules. Generally, microcapsules refer to particles ranging from 0.2 to 5000 µm, while nanocapsules are smaller than 0.2 µm. Encapsulation in the nanoscale can be used to elaborate nanodispersions with greater efficiency to be used in the beverage industry, as well as for the development of packaging materials.

Furthermore, encapsulation efficiency, size, density, and stability vary according to the encapsulating process and wall material, but factors such as speed homogenization, pH, addition of mono or divalent ions (NaCl, MgCl2), and ionic strength must be care-fully controlled to optimize the microcapsules’ properties. In addition, the microcapsule structure for single-core or multicore particles with single or multiwall structures may vary within the manufacture process.

Therefore, advances in microencapsulation involve the development of resistant and safety biopolymers that constitute a wall structure capable of preventing undesired core leakage and fulfils food industry requirements in relation to the desired application while complying with regulatory and safety aspects. Other consideration is the lack or inconsistency of the safety and toxicity studies related to the use of microcapsules; therefore, further studies must be conducted to support the legislation and safety regulations of microcapsules for food consumption”.

Reviewer 2 Report

This manuscript reviewed microencapsulation technology in food application. The advantage of microencapsulation was summaries by different food matrices. The author attempt to focus on the research from application ends rather than technical ends. Overall, this manuscript has summarized the most important manuscript recently, but there are some major concerns before publication.

First, there are lots of grammarly and typos, which need to be improved in revision. Please see details below;

Second, as a review paper, the author fails to provide a perspective for this scope,  please add at least one paragraph in your paper about the perspective of microencapsulation in food application;

Third, the author only provides few figures and a table for assistant illustration. More information need to be summarized as tables and figures.

Line 50 not controlling release during shelf life, in most cases, control release in vivo.

Line 14 de to the

Line 55, 204, 242, 280 yoghurt to yogurt

Table 1 liophilization  to  lyophilization

Table 1 Add a column for representative references

Table 1 Good option for temperature sensitive compounds to ‘a good option for temperature-sensitive compounds’

Line 64-71, hard to understand, please revise.

Line 81-84 It’s not a sentence

Line 94-95, delete among and provide a reference

Line 151 such is to such as

Line 155 need a reference

Line 156 high-pressure treatment, can accelerate oxidation processes in meat 156 products to high-pressure treatment can accelerate oxidation processes in meat 156 products

Line 158 such is to such as

Line 163 Similar to A similar

Line  182 need a reference

Line 183 EPA full name is Eicosapentaenoic acid

Line 193 delete as

Line 238, 262 fulfil to fulfill

Line 336 consume to consuming

Line 376 compound to compounds

Author Response

Response to Reviewer 2 Comments

We are very grateful for the useful suggestions raised by the reviewers and have revised the work as requested. The changes have been highlighted in red in the revised manuscript. We specify the changes below.

The review molecules-1600371 « The role of microencapsulation in food application» provided an overview of the state of art of the microencapsulation in the food industry.

Point 1. First, there are lots of grammarly and typos, which need to be improved in revision. Please see details below;

Response 1. In addition to the corrections detailed below, the final version of the manuscript was sent for language correction.

Point 2. Second, as a review paper, the author fails to provide a perspective for this scope,  please add at least one paragraph in your paper about the perspective of microencapsulation in food application;

Response 2. Excellent observation, we totally agree with it. For this reason, we added section number 6, where the perspectives are discussed. The added paragraph is attached below.

“6.       PERSPECTIVES OF MICROENCAPSULATION IN FOODS

As mentioned in this review, microencapsulation constitutes a great area to develop solutions in the food industry by incorporation of active and functional ingredients within the food matrix. Microcapsules not only offer protection and enhance bioavailability and stability of bioactive substances and natural dyes, but also improve flavor by masking the taste of fortified foods with added vitamins and minerals, and prevent their interaction with other ingredients, thus facilitating their delivery. However, some constrains may be taken to verify the quality and stability of microencapsulated ingredients such as cost, encapsulation efficiency, water solubility, release rate, particle size, taste, and microscopic structure to avoid premature release, among other undesired outcomes.

Palatability plays a role for the inclusion of microcapsules in the food matrices, as they may be rejected by consumers, especially for large-sized particles or due to variations in consistency and texture. One of the key points includes the compatibility of the food matrix that must be capable of masking the presence of large-sized microcapsules. Generally, microcapsules refer to particles ranging from 0.2 to 5000 µm, while nanocapsules are smaller than 0.2 µm. Encapsulation in the nanoscale can be used to elaborate nanodispersions with greater efficiency to be used in the beverage industry, as well as for the development of packaging materials.

Furthermore, encapsulation efficiency, size, density, and stability vary according to the encapsulating process and wall material, but factors such as speed homogenization, pH, addition of mono or divalent ions (NaCl, MgCl2), and ionic strength must be care-fully controlled to optimize the microcapsules’ properties. In addition, the microcapsule structure for single-core or multicore particles with single or multiwall structures may vary within the manufacture process.

Therefore, advances in microencapsulation involve the development of resistant and safety biopolymers that constitute a wall structure capable of preventing undesired core leakage and fulfils food industry requirements in relation to the desired application while complying with regulatory and safety aspects. Other consideration is the lack or inconsistency of the safety and toxicity studies related to the use of microcapsules; therefore, further studies must be conducted to support the legislation and safety regulations of microcapsules for food consumption”.

Point 3. Third, the author only provides few figures and a table for assistant illustration. More information need to be summarized as tables and figures.

Response 3. More information was included in the tables. However, due to the type of information, we did not find another grouping of the topic that could be captured in a table or figure.

Point 4. Line 50 not controlling release during shelf life, in most cases, control release in vivo.

Response 4. The statement was changed to: “…in other cases, microcapsules promote the product release in vivo during digestion”.

Point 5. Line 14 de to the

Response 5. Word was corrected

Point 6. Line 55, 204, 242, 280 yoghurt to yogurt

Response 6. Words were corrected

Point 7. Table 1 liophilization  to  lyophilization

Response 7. Word was corrected

Point 8. Table 1 Add a column for representative references

Response 8. We added a column with references

Point 9. Table 1 Good option for temperature sensitive compounds to ‘a good option for temperature-sensitive compounds’

Response 9. The statement was modified according to the suggestion

Point 10. Line 64-71, hard to understand, please revise.

Response 10. The paragraph was modified to read as follows: “Microcapsules in foods have various functionalities, which are summarized throughout this review of publications from the last 6 years (2015-2021) listed in data-bases such as SCOPUS, PubMed, and Web of Science. The search included words such as "microencapsulation" plus "meat", "milk", "bread", and "juice", among other foods con-sidered in the review. Articles were considered in which microcapsules were added during food preparation and presented accordingly with their specific functionality or application as a preservative, functional food development, sensory improvement, fat substitution, and/or others, which will be broken down throughout the review (Figure 1).”

Point 11. Line 81-84 It’s not a sentence

Response 11. The entire paragraph was rewritten to read as follows:

“Among the most commonly encapsulated preservatives are polyphenols (such as fla-vonoids and tannins capable of exerting antioxidant activity and inhibiting microbial growth); organic acids (which can change the pH); essential oil extracts derived from herbs, spices, onion, garlic, and fruits (set of a variety of compounds including phenolics and organosulfur compounds with antioxidant and antimicrobial capacity); bacteriocins (peptides with antimicrobial capacity); and even phage (viruses that infect bacteria). Compounds with antioxidant capacity prevent or retard the formation of free radicals and, therefore, oxidation reactions; meanwhile, antimicrobials, through various mech-anisms, are capable of disrupting membranes or cell walls and dissipating the proton motive force, inhibiting microbial growth [23]. It should be mentioned that many of the extracts with relevant properties are not soluble or compatible with the matrix to which they are to be added; so, microencapsulation may be relevant for their application”

Point 12: Line 94-95, delete among and provide a reference

Response 12. The word “among” was deleted and a reference about it was added

Point 13: Line 151 such is to such as

Response 13: Word was corrected

Point 14: Line 155 need a reference

Response 14: The idea was corrected, and include an example with reference

Point 15. Line 156 high-pressure treatment, can accelerate oxidation processes in meat 156 products to high-pressure treatment can accelerate oxidation processes in meat 156 products

Response 15. The statement was corrected

Point 16. Line 158 such is to such as

Response 16: Word was corrected

Point 17. Line 163 Similar to A similar

Response 17: Word was corrected

Point 18. Line  182 need a reference

Response 18: Reference was added

Point 19. Line 183 EPA full name is Eicosapentaenoic acid

Response 19. EPA was deleted and “Eicosapentaenoic acid” was added.

Point 20. Line 193 delete as

Response 20. Word was deleted

Point 21. Line 238, 262 fulfil to fulfill

Response 21. Word was corrected

Point 22. Line 336 consume to consuming

Response 22. Word was corrected

Point 23. Line 376 compound to compounds

Response 23. Word was corrected

Round 2

Reviewer 2 Report

This version is good for publication.